# Area-Wide Elimination of Subterranean Termite Colonies Using a Novaluron Bait

**DOI:** 10.3390/insects12030192

**Published:** 2021-02-24

**Authors:** Phillip Shults, Steven Richardson, Pierre-Andre Eyer, Madeleine Chura, Heather Barreda, Robert W. Davis, Edward L. Vargo

**Affiliations:** 1Department of Entomology, 2143 TAMU, Texas A&M University, College Station, TX 77843, USA; sric119@lsu.edu (S.R.); pieyer@live.fr (P.-A.E.); MChura@scj.com (M.C.); heatherabarreda@gmail.com (H.B.); ed.vargo@tamu.edu (E.L.V.); 2BASF Professional & Specialty Solutions, 26 Davis Drive, Research Triangle Park, NC 27709, USA; robert.davis@basf.com

**Keywords:** *Reticulitermes flavipes*, Trelona^®^, microsatellite, integrated pest management (IPM), colony elimination

## Abstract

**Simple Summary:**

Subterranean termites cause damage to man-made structures around the world and are continuing to invade new areas. Current practices for controlling termites generally target a single colony as workers tunnel near these structures, and although they are effective in most instances, they never reduce the overall termite pressure in the surrounding area. An area-wide approach to pest management could offer a way of controlling termites at the population level. By eliminating all or most of the colonies within a given area, the threat of infestation decreases. We tracked individual termite colonies over time, before and after the introduction of termite baits, to assess how long these colonies remained active to determine if a termite-free area could be maintained with continued baiting. This baiting approach was successful in significantly reducing the overall termite population within a baited area.

**Abstract:**

We investigated the use of termite baiting, a proven system of targeted colony elimination, in an overall area-wide control strategy against subterranean termites. At two field sites, we used microsatellite markers to estimate the total number of *Reticulitermes* colonies, their spatial partitioning, and breeding structure. Termite pressure was recorded for two years before and after the introduction of Trelona^®^ (active ingredient novaluron) to a large area of one of the sites. Roughly 70% of the colonies in the treatment site that were present at the time of baiting were not found in the site within two months after the introduction of novaluron. Feeding activity of the remaining colonies subsequently ceased over time and new invading colonies were unable to establish within this site. Our study provides novel field data on the efficacy of novaluron in colony elimination of *Reticulitermes flavipes,* as well as evidence that an area-wide baiting program is feasible to maintain a termite-free area within its native range.

## 1. Introduction

The area-wide (AW) approach to integrated pest management (IPM) aims to lower overall pest pressure by focusing control at a population level, rather than by targeting individual infestations [1]. By reducing a pest population within a delimited area, the problems associated with the pest can be greatly minimized. AW-IPM has shown high levels of success against a variety of agricultural and vector pest species, including boll weevils (Curculionidae), gypsy moths (Erebidae), and several mosquito species (Culicidae) [1]. AW control is notably effective on pests that are less mobile and repopulate slowly. Thus, integrating AW approach into termite management program should be highly effective. Results from several studies using AW termite baiting studies targeting invasive species of termites look encouraging (*Coptotermes formosanus* Shiraki in New Orleans, USA and *Reticulitermes flavipes* Kollar in Santiago, Chile) [2,3,4,5]; however, more work is needed to test the viability of this control strategy against different pest species of termites, as well as evaluate the efficacy of various commercially available termite baits.

Subterranean termites are present throughout much of the world in both undisturbed and anthropogenic habitats. As decomposers of cellulose material, they provide valuable ecosystem services; however, they can also cause massive amounts of damage to human structures. In the United States alone, homeowners spend over $11 billion USD annually on subterranean termite control measures and damage repairs [6,7]. In addition, subterranean termites include highly invasive species rapidly spreading worldwide, such as *C. formosanus, C. gestroi* (Wassman) *and Reticulitermes flavipes* [8,9,10,11,12,13]. The genus *Reticulitermes* contains the most widespread subterranean termite species in the U.S., as well as some of the most damaging pest species: *R. flavipes* and *R. virginicus* Banks [14]. Colonies of these species can contain hundreds of thousands or even millions of foraging termites [15], and colony density can be as high as 300 colonies per hectare [16], although this varies geographically and by habitat type [17]. With such a high colony density, man-made structures in these areas can be subject to an immense amount of termite pressure. Therefore, AW control may be of particular relevance to reducing termite pressure within a given area. 

In general, new colonies of the subterranean termite *R. flavipes* are founded by a monogamous pair of primary reproductives. These primary reproductives disperse through flight and can populate new areas; however, newly founded colonies require years to fully mature [14]. Once established, colonies are generally stationary with workers radiating from the central nest to forage [16]. At this stage of the colony, workers are the offspring of the outbred primary reproductives, which constitutes a simple family. When the primary queen or king dies, nymphs or workers can further develop into neotenic reproductives, which can extend the colony’s life span through inbreeding (extended-family colonies) [18,19,20]. Sometimes, distinct colonies can merge into a single genetically diverse, yet cohesive, colony (mixed family) [21]. Overall, these differences in family types may greatly influence the size of the colonies, and consequently their foraging range [19,22]. Throughout most of the southeastern US, roughly 65–85% of *R. flavipes* colonies are simple families, however, the proportion of simple-family colonies to extended- or mixed-family colonies varies drastically in the northeast US and Louisiana [23]. As mentioned above, *R. flavipes* is also invasive in large regions in France, Germany, Canada, and Chile and is continuing to expand to other areas [9,24,25,26,27]. In these invasive localities, colonies may be even larger than within the native range due to low non-nestmate aggression, leading to a higher frequency of colony fusion [28,29]. In addition, these colonies are often headed by numerous (up to hundreds) neotenics, which results in highly populated colonies. With such drastic variation in the breeding structure and behavior of *R. flavipes* populations in both the native and invasive ranges, measures taken to control these termites could have varying levels of success. 

The two most commonly used methods of controlling termite infestations are treating the soil adjacent to the structure with a liquid termiticide or installing bait stations [30,31,32]. These site-specific control measures attempt to protect the structure from invading colonies, but the overall threat of re-infestation remains as neighboring colonies can move into these vacated areas [33,34]. Termite baits are used as both curative and preventative measures against subterranean termites and have been shown to eliminate colonies [30,35,36]. The active ingredients of most termite baits are chitin synthesis inhibitors (CSI), which cause mortality in the workers when they return to the nest to molt and spreads through the colony through trophallaxis. Over time, the colony collapses and is presumed eliminated [37]. On average, colony elimination occurs after 3–6 months of bait consumption with a 90–100% efficacy rate [38]. As effective as termite baiting is, this strategy is usually used to target termite colonies associated with a structure without addressing the overall pest pressure in a larger area. Integrating baiting into an AW management program aims at eliminating most of the colonies within a large area, greatly reducing the threat of termites attacking a structure [37]. This approach has shown that it is possible to significantly decrease overall termite pressure over multiple city blocks [3,4,39,40], but most of the AW-IPM baiting programs have been centered on invasive populations of *C. formosanus* in the USA and *R. flavipes* in Chile. As native populations of subterranean termites have increased genetic diversity, face locally adapted pressures, and encounter a potentially higher availability of suitable habitats, it is unclear to what extent AW baiting can be effective. Additionally, nearly 80% of all field termite baiting trials have been conducted using either hexaflumuron or noviflumuron, highlighting the need for testing other active ingredients to determine if these CSI baits can achieve similar results [38]. 

In this study, we tested the viability of using Trelona^®^ Compressed Termite Bait (active ingredient novaluron) and the Advance^®^ Termite Baiting System (ATBS) to target and reduce the total number of subterranean termite colonies within a given area. For two years, we tracked the total number of *Reticulitermes* colonies and mapped their spatial partitioning to determine the baseline termite pressure in a delimited area. Colony differentiation was genetically assessed using workers collected from the bait stations. Novaluron was then introduced to a large section of the treatment site, and colonies were tracked for two additional years. We estimated the time elapsed before the treated colonies collapsed as well as consumption of the CSI bait. Of the original termite colonies identified in the treatment site, none remained after baits were applied and new colonies invading the site were not active for long. This study provides novel field data for colony elimination and successful AW control using novaluron.

## 2. Materials and Methods

### 2.1. Field Site and Collections

This study was conducted in College Station, Texas at field sites located near the Rollins Urban and Structural Entomology Facility. In February 2016, 200 ATBS^®^ termite stations were installed at two field sites (100 per site) with 5 m between each station (Figure 1 and Figure 2). In each site, approximately half of the stations were placed within a wooded area and the other half were placed in an adjacent pasture. During initial installation and for the first 27 months of the study, all stations within these two sites contained a set of two wooden monitoring blocks and blank (no active ingredient) termite inspection cartridge (Appendix A). These blank cartridges contain the same cellulose bait matrix as found in the commercially available product but without novaluron added. The wooden monitoring blocks and inspection cartridges were replaced as needed due to mold, degradation, and termite consumption. Stations were inspected every other month, and, approximately 50 individuals were collected and stored in 95% ethanol from stations found with active termites. These samples were stored at −20 °C for later genotyping and colony assignment. For each station with termite activity, four auxiliary stations were placed 2.5 m from the main station in each cardinal direction. In May of 2018, 55 stations (yellow triangle (Figure 2)) with the highest termite activity in the treatment site (~1400 m^2^ of wooded area (Figure 2)) had the wooden monitoring blocks and inspection cartridges replaced with two novaluron bait cartridges (Appendix A). The remaining stations in the treatment site and all of the stations in the control site were left untreated (wooden monitoring blocks and blank inspection cartridges). Inspections were carried out monthly for the remaining 17 months of the study. Feeding on novaluron bait was determined by visual inspections to estimate the amount of bait consumed. The mean number of active stations per inspection at each site was compared using a generalized linear model (GLM) with the interaction between site and pre-/post-treatment as fixed effects, followed by Tukey’s HSD in JMP 14.0 (SAS software, Cary, NC, USA).

### 2.2. DNA Extraction, Genotyping & Sequencing

Total DNA was extracted from eight workers from each collection sample following a modified Gentra PureGene protocol (Gentra Systems, Inc., Minneapolis, MN, USA). Two highly polymorphic microsatellite markers (*Rf21-1* and *Rf24-2*) were amplified for each individual following the method of Vargo 2000 [41]. The *Rf21-1* primer set was fluorescently labeled using 6-FAM dye and the *Rf24-4* was labeled using NED dye. DNA amplifications were performed in a volume of 15 µL including 0.15 µL of MyTaq™ HS DNA polymerase (Bioline, Cincinnatri, OH, USA), 5 µL of MyTaq™ 5× reaction buffer (Bioline, Cincinnatri, OH, USA), 0.50 µL of each primer, and 1.0 µL of the DNA template. PCR was carried out using a Bio-Rad thermocycler T100 (Bio-Rad, Hercules, CA, USA) using the following program: initial denaturation step at 94 °C (50 s) followed by 35 cycles at 94 °C (50 s), 55 °C (2 min), and 72 °C (2 min), with a final extension step at 72 °C (5 min). PCR products were visualized on an ABI 3500 genetic analyzer using a LIZ500 internal standard (Applied Biosystems, Foster City, CA, USA). Allele labeling was performed using Geneious software v.9.1 [42]. A portion of the 16S gene region was sequenced from an individual worker for each inferred colony collected in the first year to determine species. Further species differentiation was assessed using species-specific microsatellite alleles on the *Rf21-1* and *Rf24-2* markers for three years (described below).

### 2.3. Colony Breeding Structure and Colony Differentiation

To determine if workers from different collection samples originated from the same or different colonies, we compared the genotypes of workers between each pair of collection samples. *F*_ST_-values were generated for all pairs of samples and the genotypic frequencies were compared using a log-likelihood (G)-based test of differentiation in Genepop version 4.7.0 [43]. Workers from two different samples were considered to belong to the same colony when the two samples exhibited low pairwise *F*_ST_ value and when allelic frequencies did not differ significantly from the expected based on the G-test (*p* < 0.05) (Appendix A). Within each year, all of the collection samples were analyzed and assigned to colonies. Collection samples belonging to a unique colony were combined within a given year. The colonies (i.e., grouped collection samples) were subsequently compared across the four years (Appendix A). These analyses were performed for each site separately and allowed us to track the fate of individual colonies throughout the study. Colonies with a mean *F*_IC_ value across both loci of ≤−0.1 were considered to be simple-family colonies [44,45].

## 3. Results

Over the length of the study, a total of 298 termite samples were collected (Figure 3a), which resulted in 2673 workers genotyped from both sites. The mean number of active stations per inspection pre-baiting was not significantly different between the control site and the treatment site, with 5.9 and 5.4 active stations, respectively (Tukey’s HSD, *p* = 0.9642) (Figure 3b). There was also no significant difference in the mean number of active stations in the control site pre- and post-baiting of the treatment site (Tukey’s HSD, *p* = 0.6677). The mean number of active stations was significantly lower in the treatment site after novaluron was introduced (Tukey’s HSD, *p* = 0.0027) (Figure 3b). The overwhelming majority of termites collected came from stations within the wooded areas (Figure 1 and Figure 2) with only 6.3% collected from stations within the pastures.

The 16S sequencing revealed that both *R. flavipes* and *R. virginicus* were present in both sites. The markers *Rf21-1* and *Rf24-2* were highly polymorphic in *R. flavipes* (21 and 28 alleles respectively), and thus sufficient to determine colony differentiation. However, the allelic diversity of these two markers was much lower for *R. virginicus* (4 alleles per marker). Interestingly, the four alleles found in this species were not found in *R. flavipes*. Although this weak polymorphism hampers proper colony differentiation on *R. virginicus*, these species-specific alleles allowed for the separation of the two termite species based on the microsatellite markers only. Of the samples collected, roughly 30% were found to be *R*. *virginicus*. Over the total length of the study, 32 *R. flavipes* colonies were identified; 17 from the control site and 15 from the treatment site (Figure 1 and Figure 2). The mean *F*_IC_ of each colony did not change drastically over time (Appendix A); however, both simple and extended family colonies were present (Table 1). No mixed-family colonies were identified in either site.

When untreated, each site hosted between 8–14 colonies within a given year, which equates to roughly 55–100 colonies per hectare. The colonies within a site changed slightly over time, but tended to only occupy a small area (1–2 stations) with little movement within the respective site. In the treatment site, 13 colonies were identified before the CSI baits were introduced (Figure 2). Two of these colonies were not present at the time of baiting, and of the 11 remaining colonies, seven were no longer active within two months of introducing novaluron. The other three colonies were only collected in auxiliary stations and were not observed feeding on novaluron until approximately 8–10 months after baiting, at which point they were presumably eliminated. No termites were collected in the treatment site from June 2019–October 2019. In comparison, 12 colonies were identified in the control site before baiting, and of those, 7 were still present in the site toward the end of the study in October 2019 (Figure 4). In the treatment site, the mean amount of novaluron bait consumed per colony was ~78.0 g (just over half of one termite bait cartridge) though as little as 10.0–30.0 g of bait was required to eliminate the foraging activity of some colonies (Table 2).

## 4. Discussion

Area-wide pest control aims to eliminate pest populations before they become a problem and the data presented here is strong evidence that this can be done against subterranean termites of the genus *Reticulitermes* using novaluron. This study also illustrates the substantial termite pressure man-made structures may face, especially those adjacent to a wooded area. By reducing the number of colonies in these natural areas, the overall threat of structural infestation is significantly reduced. The efficacy of CSI baits for the control of single colonies of subterranean termites is well established; however, their use in AW control has only been previously demonstrated using noviflumuron. Using novaluron in an AW baiting program, both established colonies and new invading colonies were presumably eliminated. Before baiting, the termite activity was constant in the treatment site, with many colonies spanning multiple years. Some natural turnover was observed as 2–3 colonies were replaced, but overall, this site was able to support 8–12 *R. flavipes* colonies. Consequently, we can hypothesize that it will likely take roughly 3–6 years for the treatment site to return to the colony density pre-treatment. In general, *R*. *flavipes* colonies occupied a small foraging range with minimal overlap between colonies, which is consistent with what was found in previous studies [16,44,46]. In a few instances, two colonies were collected from the same station at different times, and this may be due to the utilization of another colony’s pre-formed foraging tunnels [36,47,48]. 

Here, approximately 70% of the *R. flavipes* colonies collected were simple families and the remaining 30% were extended families (Table 1). This proportion of colony types is common in the southern US [23,49]. Extended-family colonies tend to have an increased foraging range, and in general, we found the colonies occupying multiple stations tended to be extended families (Figure 1 and Figure 2; Table 1). This increased foraging could make extended-family termite colonies more susceptible to AW baiting as the workers are more likely to encounter a bait station. In *R. flavipes* this may be of particular interest in the invasive populations in France, where all colonies are extremely large extended and mixed families [23,50]. Whereas the increase in foraging within these populations might make baiting effective, the increase in worker production may also allow these colonies to compensate for worker death caused by CSI baits. Additionally, some species of *Reticulitermes*, including *R. virginicus,* have a complex breeding system known as asexual queen succession [51,52]. In species that employ an asexual queen succession system, the primary queen can reproduce parthenogenically to produce neotenic queens. After the primary dies, these secondary queens can mate with the primary king and produce offspring. These now effectively polygyne colonies can have drastically increased growth and reproductive output, similar to extended- or mixed-family colonies [53,54], however, asexual queen succession colonies avoid inbreeding and thus maintain genetic diversity in the colony [55]. While we did not observe differences in the efficacy of baiting simple and extended-family colonies, there are many different colony breeding systems within termites [56], and these have the potential of reducing the effectiveness of AW-IPM. 

The amount of bait fed upon by each colony was highly variable, ranging from 10.0–250.0 g (Table 1), though as little as 1 g has been shown to cause colony collapse [57]. Inactivity of roughly 70% of the colonies in the treatment site occurred within 4–8 weeks after baiting. The time to elimination is dependent on how soon a colony starts to feed on the CSI baits [58,59], and as these colonies were active in the stations at the time novaluron was introduced, it is presumed that feeding started immediately. Decreased time to colony elimination has also been shown in other baiting systems that induce immediate feeding (i.e. above-ground stations and fluidized baits) [60,61]. Interestingly, these baits may also weaken the colony to the point they suffer and collapse from alternative and opportunistic pathogenic agents naturally present in the soil [62,63,64]. In addition to the baiting had on the original colonies in the treatment site, new colonies collected after the baits were introduced had a severely truncated presence compared to the new colonies collected in the control site (Figure 4). Additionally, subsequent termite colonies may utilize the existing tunnels of previous colonies potentially leading them to bait stations. Once all of the colonies within an area have been eliminated, reinvasion by new colonies will occur; however, with the development of more durable baits and yearly monitoring by pest management professionals, presumably, termite-free areas can be maintained as long as the baits are present [37].

## Figures and Tables

**Figure 1 insects-12-00192-f001:**
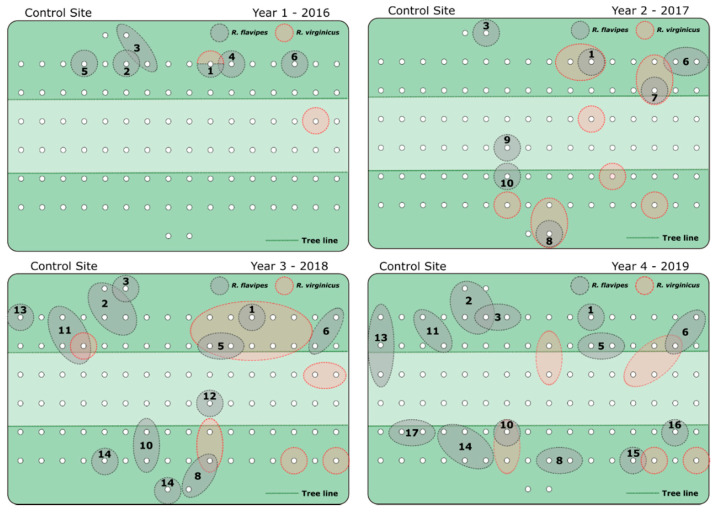
Colonies in the untreated control site by year. Individual colonies were assigned to a number within each site. White circles represent an individual ATBS^®^ station (auxiliary stations not shown). Stations within the boundary of a colony indicate instances where each colony was collected (foraging range for that specific colony). Dark green areas indicate wooded areas with the light green being open pasture.

**Figure 2 insects-12-00192-f002:**
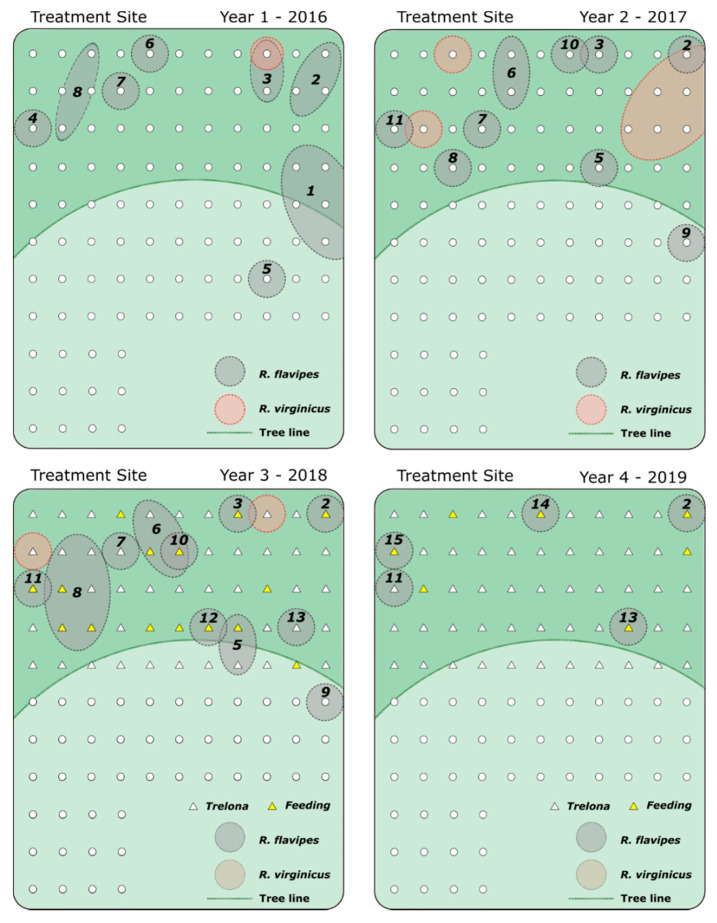
Colonies in the treatment site by year. Individual colonies were assigned to a number within each site. White circles represent an individual ATBS^®^ station (auxiliary stations not shown). Stations within the boundary of a colony indicate instances where each colony was collected (foraging range for that specific colony). In 2018, stations within the wooded area were each loaded with two novaluron bait cartridges (triangles) and cartridge feeding within the stations is denoted in yellow. Dark green areas indicate the wooded areas with the light green being open pasture.

**Figure 3 insects-12-00192-f003:**
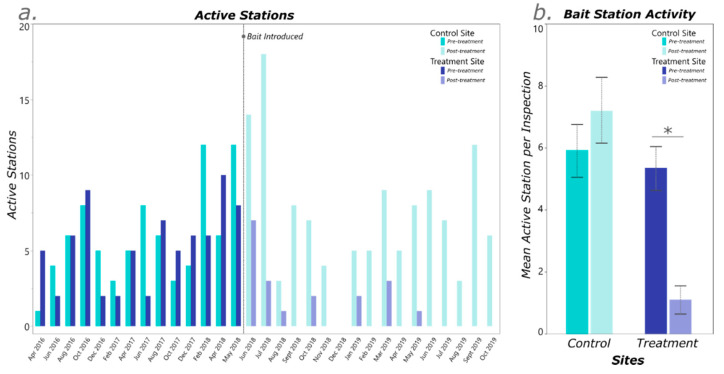
(**a**) Number of stations with termite activity during each inspection for the control and treatment site. (**b**) The mean number of monthly active stations pre- and post-treatment per site. Values found to be significantly different are denoted with an * (GLM with Tukey’s, *p* < 0.05).

**Figure 4 insects-12-00192-f004:**
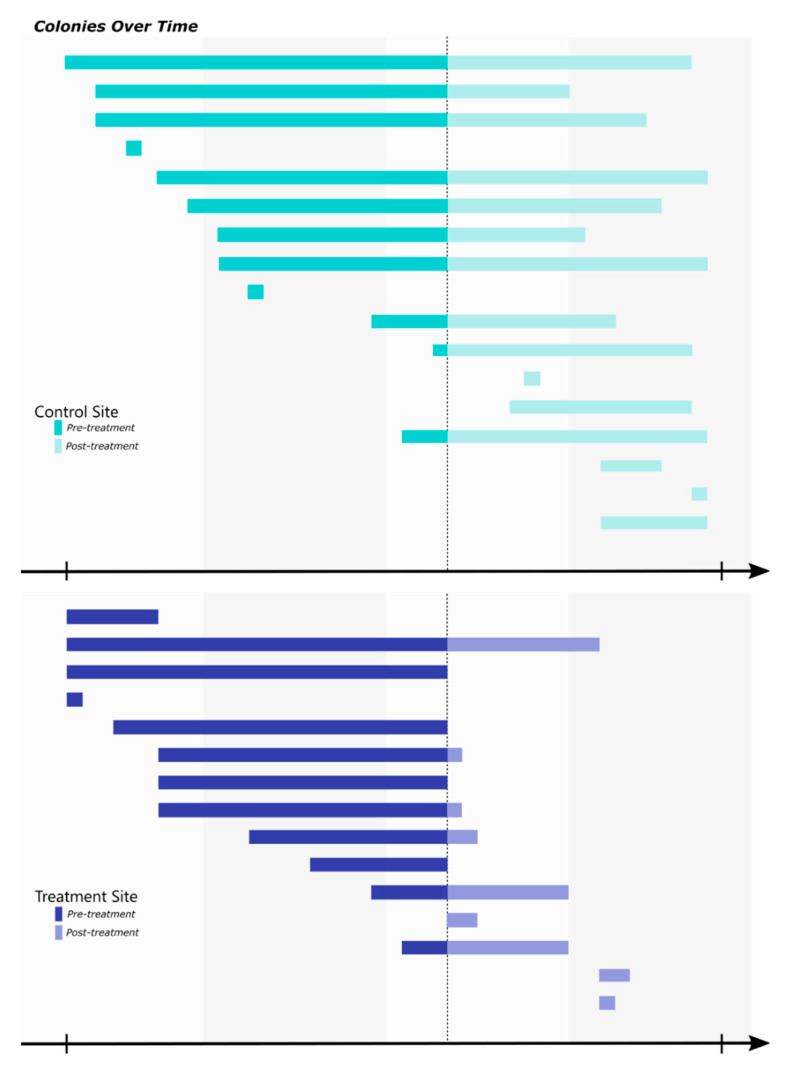
The amount of time each colony was active in the field sites. Each bar represents an individual colony and the length of the bar indicates the period of time in which it was collected.

**Table 1 insects-12-00192-t001:** The mean inbreeding coefficient (*F*_IC_) for each colony identified for each site.

Treatment Site				Control Site			
Colony	Workers Genotyped	*F* _IC_	Family Type	Colony	Workers Genotyped	*F* _IC_	Family Type
1	87	−0.3338	Simple	1	57	0.3031	Extended
2	95	−0.2628	Simple	2	123	−0.4620	Simple
3	70	−0.3608	Simple	3	84	−0.3088	Simple
4	10	0.5610	Extended	4	9	−0.2948	Simple
5	86	−0.3590	Simple	5	87	−0.3536	Simple
6	89	0.2324	Extended	6	177	0.4706	Extended
7	30	−0.3126	Simple	7	7	−0.5385	Simple
8	78	0.2104	Extended	8	116	−0.2485	Simple
9	29	−0.3764	Simple	9	10	0.2070	Extended
10	22	−0.3057	Simple	10	107	−0.2718	Simple
11	47	0.0439	Extended	11	89	−0.4527	Simple
12	10	−0.2289	Simple	12	7	−0.0366	Extended
13	29	−0.0950	Simple	13	56	−0.2535	Simple
14	19	−0.1060	Simple	14	101	0.4979	Extended
15	8	0.3058	Extended	15	30	−0.2353	Simple
v	155	−0.1764	Simple	16	9	0.1818	Extended
				17	40	−0.2966	Simple
				v	700	−0.2772	Simple

**Table 2 insects-12-00192-t002:** Estimation of total amount of bait consumed per colony in the treatment site.

Treatment Site		
Colony ID	Amount of Bait Consumed (g)	Last Seen in Site
1	-	Oct-16
2	60	May-19
3	90	May-18
4	-	May-16
5	250	May-18
6	60	Jun-18
7	30	May-18
8	210	Jun-18
9	30	Jul-18
10	30	May-18
11	10	Jan-19
12	60	Jul-18
13	120	Jan-19
14	60	May-19
15	30	Mar-19

## Data Availability

The data presented in this study are available in Appendix A.

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
