# Peer review of "Area-Wide Elimination of Subterranean Termite Colonies Using a Novaluron Bait"

_insects, 2021, doi:10.3390/insects12030192_

Round 1

Reviewer 1 Report

Overall concept is good needs work:

  • wooded area- why type of wood/forest needs description
  • pasture area –why and what type if field (i.e. vegetation)
  • Authors indicated that other AW worked for above ground insects but termites are below ground???
  • Fig 1 need to indicate # for colonies ID
  • Fig 2 Why do the pasture site have no triangles? Very low colonies in pastures less than 15% why did they use them?
  • Fig 3 Samples collected monthly but not represented on the graph- why?
  • No materials and methods for bait weights but presented in results. How were the weights determined? Major oversight by authors.
  • No justification for AW but in one in theory not practical in real world. Discussion should include pros and cons of the AW. Such as who pays for the install and inspection of the bait stations –homeowners? No discussion on “buys –in” by homeowners. Different pest control companies involved and they do things differently.  No indication of how they would implement AW program.
  • Not a fan of “colony elimination” I think colony suppression is more likely.

Author Response

Author response to reviewer:

1) wooded area- why type of wood/forest needs description

Author response: College Station is located in the post oak savannah region of Texas, thus most of the forested areas are predominantly oaks and most of the open fields are grasslands.

2) pasture area –why and what type if field (i.e. vegetation)

Author response: See comment above. We can add this information if need be, but we feel that this information is accessible to anyone interested. One potential change that may help is to change “pasture” to “grass field” in line 110.

3) Authors indicated that other AW worked for above ground insects but termites are below ground???

Author response: This statement was to introduce the concept of area-wide pest control and show that it has been successful in other systems. We dedicated the rest of the manuscript to assessing if this type of control strategy could work against termites.

4) Fig 1 need to indicate # for colonies ID

Author response: This has been added to line 135.

5) Fig 2 Why do the pasture site have no triangles? Very low colonies in pastures less than 15% why did they use them?

Author response: There was very little activity in these stations and to conserve resources and the time and energy of the personnel involved, we decided to only bait the area with high termite activity. To the second point, when these stations were installed, we did not know where the termite activity would be or if any colonies would forage into the field.

6) Fig 3 Samples collected monthly but not represented on the graph- why?

Author response: Originally, the stations were checked every other month (line 116). After the baits were installed at the treatment site, inspections were carried out monthly (line 123).

7) No materials and methods for bait weights but presented in results. How were the weights determined? Major oversight by authors.

Author response: This sentence was changed to clarify how the weights were determined (lines 124-126).

8) No justification for AW but in one in theory not practical in real world. Discussion should include pros and cons of the AW. Such as who pays for the install and inspection of the bait stations –homeowners? No discussion on “buys –in” by homeowners. Different pest control companies involved and they do things differently. No indication of how they would implement AW program.

Author response: The goal of this work was to show the versatility of termite baiting. Much of the work on baiting has focused on the protection of a single structure from a single infestation, but here we show that where baits are maintained, there is very little termite activity. An individual homeowner or pest control operator would not need to go to the extent to which we did. As shown in figure 2, the addition of 4-6 bait stations at the tree line was enough to eliminate at least 3 colonies that would pose a potential threat to a structure. There are so many variables to consider in the implementation of an AW baiting program that we purposefully excluded this from our discussion as it would be mainly conjecture. Additionally, reference 37 talks at length about how to successfully implement and streamline an AW termite baiting program.

9) Not a fan of “colony elimination” I think colony suppression is more likely.

Author response: The term “colony elimination” has been reduce throughout.

Reviewer 2 Report

This is a very nice study and well presented. It is great to see such a well designed study on area-wide control of termites, especially given a proper control group. I know it is too much to ask given the flagrant use of the word elimination by competing bait companies, but I still will suggest you eliminate eliminate where every you possibly can. You did not show area-wide elimination, or as you state on line 237 pest populations were eliminated (though in a sense you are correct). Try to take yourselves and the termite community to a higher level.  I fact, you almost hide a most important understanding of your results when you state that within as little as 3yrs pretreatment termite densities can be expected to return--after an "elimination".  Such a finding is important to highlight and and it will distance yourselves from an industry misperception that continues to be a plague on the science.  Line 63-65 is incorrect in stating that new colonies are only formed from monogamous pairs where primary reproductives originate from different colonies.  This is simply not true and all I can figure is it is used to support the genetic underpinnings of your work (though you don't say this). In fact you know this is not true as stated in lines 255-256 where you found both colonies as you describe in Line 63-65 and the sometimes more common colonies (at least in disturbed urban areas) that are 'extended families'.  And why would you say paired primary reproductives are always from different colonies?  line 87 add attempt before to protect.  References #3 and #38 are identical.

Author Response

Author Response: We appreciate the reviewers comment and suggestions regarding the manuscript, and below are our responses.

I know it is too much to ask given the flagrant use of the word elimination by competing bait companies, but I still will suggest you eliminate eliminate where every you possibly can. You did not show area-wide elimination, or as you state on line 237 pest populations were eliminated (though in a sense you are correct). Try to take yourselves and the termite community to a higher level.  I fact, you almost hide a most important understanding of your results when you state that within as little as 3yrs pretreatment termite densities can be expected to return--after an "elimination".  Such a finding is important to highlight and it will distance yourselves from an industry misperception that continues to be a plague on the science. 

Author Response: In regards to the use of the word elimination, we agree that the populations were not eliminated and have changed these instances to say “population reduction” (Lines: 15 and 209). We have also reduced the number of times eliminate or elimination were used (Lines: 21, 22, 81, 93, 101, 191, 234, 236, 242, 243). We tried to remove this phrase where possible, but when referencing the work of others, this was difficult. We reduced its usage by about half but we request that elimination stays in the title and keywords for increased searchability of this work. What might be of interest moving forward is to sample this site years after the baits are removed to see if any of the colonies truly did escape elimination.

Line 63-65 is incorrect in stating that new colonies are only formed from monogamous pairs where primary reproductives originate from different colonies.  This is simply not true and all I can figure is it is used to support the genetic underpinnings of your work (though you don't say this). In fact you know this is not true as stated in lines 255-256 where you found both colonies as you describe in Line 63-65 and the sometimes more common colonies (at least in disturbed urban areas) that are 'extended families'.  And why would you say paired primary reproductives are always from different colonies?

Author Response: Lines 56-57: It was not our intention to insinuate that outbred monogamous pairs of alates are the only way in which colonies are formed and so we have added a precursory statement of “In general” to this sentence. This particular sentence is specifically referring to colony foundation, not formation, but the reviewer is correct in that they do not necessarily have to be from different or completely unrelated colonies. Colonies can be founded by siblings, relatives, more than 2 reproductives, or even two females, and though not ideal, offer a better chance of survival than not pairing at all. The phrase “originating from different colonies” has been removed.

line 87 add attempt before to protect. 

Author Response: Line 75: This correction was made.

References #3 and #38 are identical.

Author Response: Line 334: This repeated reference was eliminated.

Reviewer 3 Report

The authors examine the use of novaluron bait in eliminating termite colonies within a field site/area. The authors identify termite species and characterize individual populations using monitoring devices for termite collection and genetics. This study is very well done and the field experiments well thought out. The use of baits with an area wide approach is certainly a promising way to control termites that really hasn’t been studied enough. I am guessing one hurdle to this approach is in getting pest control operators to get on board since it would involve getting funding from a community rather than a single homeowner…but that is outside the scope of the manuscript here.

I have made a few, mostly editorial, suggestions as follows:

Lines 14 & 22: Delete (AW) from simple summary and abstract

Lines 41- 42: make two sentences, end first at “and repopulate slowly.” Begin next sentence with “Thus, integrating an AW…”

Lines 92-93: In this discussion on mode of action for baiting, trophallaxis should be mentioned somewhere as this is how the toxicant would be spread throughout the colony.

Line 98: “infesting a structure” should probably be changed to “threat of termite attack or re-infestation”

Line 111: I don’t believe colony of origin was determined here. To me it seems like results presented are just showing similarity or differentiation between all of the colonies identified rather than dispersal from a colony to other colonies which is what is implied by the phrase “colony of origin”.

Line 128: can you give more information regarding the composition of the “no active ingredient blank” cartridge?

Lines 129-130: delete sentence, “The wooden monitoring blocks….in the top of the respective station.” As this is already described and is shown in figure S1

Line 133: station should be stations

Line 136: triangle should be triangles

Line 137: delete the word level

Lines 142-144: This sentence is ambiguous. Do you mean both feeding on novaluron bait was recorded and visual inspections were done? If so, state how feeding on novaluron bait was recorded. Or, do you mean Feeding on novaluron bait was determined by visual inspections to estimate the amount of bait consumed?

Line 144: change were compared to was compared

Notes on Figures:

-Online the figures look alright, but when printed it is difficult to see differences in colors. I would consider changing them a bit just to make them stand out better. Not essential, just an observation.

-State what the numbers represent in figure 1 and 2 legends.

Line 163: give PCR conditions used

Line 167: delete the word identity

Items for discussion:

Did you see any variation in treatment of monitoring between the two species? Some research has suggested that R. virginicus is sensitive to disruption, even the act of checking baiting stations can be an issue.

Lines 267-274: How does colony breeding system impact the colony data results presented in this study?

Author Response

Author Response: We appreciate the reviewers comment and suggestions regarding the manuscript, and below are our responses.

The authors examine the use of novaluron bait in eliminating termite colonies within a field site/area. The authors identify termite species and characterize individual populations using monitoring devices for termite collection and genetics. This study is very well done and the field experiments well thought out. The use of baits with an area wide approach is certainly a promising way to control termites that really hasn’t been studied enough. I am guessing one hurdle to this approach is in getting pest control operators to get on board since it would involve getting funding from a community rather than a single homeowner…but that is outside the scope of the manuscript here.

I have made a few, mostly editorial, suggestions as follows:

Lines 14 & 22: Delete (AW) from simple summary and abstract

Author Response: This correction has been made.

Lines 41- 42: make two sentences, end first at “and repopulate slowly.” Begin next sentence with “Thus, integrating an AW…”

Author Response: This correction has been made.

Lines 92-93: In this discussion on mode of action for baiting, trophallaxis should be mentioned somewhere as this is how the toxicant would be spread throughout the colony.

Author Response: The sentence now reads: “…which cause mortality in the workers when they return to the nest to molt and spreads through the colony through trophallaxis.”

Line 98: “infesting a structure” should probably be changed to “threat of termite attack or re-infestation”

Author Response: Infesting was changed to attacking.

Line 111: I don’t believe colony of origin was determined here. To me it seems like results presented are just showing similarity or differentiation between all of the colonies identified rather than dispersal from a colony to other colonies which is what is implied by the phrase “colony of origin”.

Author Response: We have removed this sentence from lines 95-96 and have added a sentence that reads “Colony differentiation was genetically assessed using workers collected from the bait stations” (lines97-98).

Line 128: can you give more information regarding the composition of the “no active ingredient blank” cartridge?

Author Response: Additionally, information has been added to lines 111-112.

Lines 129-130: delete sentence, “The wooden monitoring blocks….in the top of the respective station.” As this is already described and is shown in figure S1

Author Response: This correction has been made.

Line 133: station should be stations

Author Response: This correction has been made.

Line 136: triangle should be triangles

Author Response: This correction has been made.

Line 137: delete the word level

Author Response: This correction has been made.

Lines 142-144: This sentence is ambiguous. Do you mean both feeding on novaluron bait was recorded and visual inspections were done? If so, state how feeding on novaluron bait was recorded. Or, do you mean Feeding on novaluron bait was determined by visual inspections to estimate the amount of bait consumed?

Author Response: We did mean the latter. We have changed this sentence to what was suggested.

Line 144: change were compared to was compared

Author Response: This correction has been made.

Line 163: give PCR conditions used

Author Response: This information was added to lines 143-145.

Line 167: delete the word identity

Author Response: This correction has been made.

Notes on Figures:

-Online the figures look alright, but when printed it is difficult to see differences in colors. I would consider changing them a bit just to make them stand out better. Not essential, just an observation.

Author Response: We have changed all the figures in the manuscript in order to improve readability and consistency. Additionally, we have combined figures 3 and 4 from the previous version into one figure.

-State what the numbers represent in figure 1 and 2 legends.

Author Response: “Individual colonies were assigned to a number within each site” was added to the figure captions.

Items for discussion:

Did you see any variation in treatment of monitoring between the two species? Some research has suggested that R. virginicus is sensitive to disruption, even the act of checking baiting stations can be an issue.

Author Response: While we don’t know the colony identity of the R. virginicus collected, in some cases we were able to collect this species from the same stations over several years. That being said, the R. virginicus did seem to be more mobile within a site. Whether this is due to us forcing them into other stations or just an artifact of an increase foraging range, it’s hard to say.

Lines 267-274: How does colony breeding system impact the colony data results presented in this study?

Author Response: In this study, there was no difference in the efficacy of baiting simple- or extended-family colonies and this has been added to line 235. We just want to highlight that this should be tested in other areas that R. flavipes occurs as the results could vary depending on the main breeding structure of a population.